# Formalising the Pathways to Life Using Assembly Spaces

**DOI:** 10.3390/e24070884

**Published:** 2022-06-27

**Authors:** Stuart M. Marshall, Douglas G. Moore, Alastair R. G. Murray, Sara I. Walker, Leroy Cronin

**Affiliations:** 1School of Chemistry, University of Glasgow, Glasgow G12 8QQ, UK; stuart.marshall@glasgow.ac.uk (S.M.M.); alastair.murray@glasgow.ac.uk (A.R.G.M.); 2BEYOND Center for Fundamental Concepts in Science, Arizona State University, Tempe, AZ 85281, USA; doug@39alpharesearch.org; 3School of Earth and Space Exploration, Arizona State University, Tempe, AZ 85281, USA

**Keywords:** complexity, information, graphs, biosignatures

## Abstract

Assembly theory (referred to in prior works as pathway assembly) has been developed to explore the extrinsic information required to distinguish a given object from a random ensemble. In prior work, we explored the key concepts relating to deconstructing an object into its irreducible parts and then evaluating the minimum number of steps required to rebuild it, allowing for the reuse of constructed sub-objects. We have also explored the application of this approach to molecules, as molecular assembly, and how molecular assembly can be inferred experimentally and used for life detection. In this article, we formalise the core assembly concepts mathematically in terms of assembly spaces and related concepts and determine bounds on the assembly index. We explore examples of constructing assembly spaces for mathematical and physical objects and propose that objects with a high assembly index can be uniquely identified as those that must have been produced using directed biological or technological processes rather than purely random processes, thereby defining a new scale of aliveness. We think this approach is needed to help identify the new physical and chemical laws needed to understand what life is, by quantifying what life does.

## 1. Introduction

In the thought experiment known as the “infinite monkey theorem”, an infinite number of monkeys, each having a typewriter, produce strings of text by hitting keys at random [1]. Given infinite resources, it can be deduced that the monkeys will produce all possible strings, including the complete works of Shakespeare. However, when constrained to the bounds of the physical universe, the likelihood that any particular text is produced by a finite number of monkeys drops rapidly with the length of the text [2]. This can also be extended to physical objects such as cars, planes, and computers, which must be constructed from a finite set of objects—just as meaningful text is constructed from a finite set of letters. Even if we were to convert nearly all matter in the universe to object-constructing monkeys, and give them the age of the universe in which to work, the probability that any monkey would construct any sufficiently complex physical object is negligible [3]. This is an entropic argument—the number of possible arrangements of the objects of a given composition increases exponentially with the object size. For example, if the number of possible play-sized strings is sufficiently large, it would be practically impossible to produce a predetermined Shakespearean string without the author. This argument implies that information external to the object itself is necessary to construct an object if it is of sufficiently high complexity [4,5]. In biology, the requisite information partly comes from DNA, the sequence of which has been acquired through progressive rounds of evolution. Although Shakespeare’s works are—in the absence of an appropriate constructor [6] (an author)—as likely to be produced as any other string of the same length, our knowledge of English grammar and syntax allows us to partition the set of possible strings, distinguishing the very small proportion that contains meaningful information.

Biological systems have access to a lot of information—genetically, epigenetically, morphologically, and metabolically—and the acquisition of that information occurs via evolutionary selection over successive cycles of replication and propagation [7]. One way to look at such systems is by comparing the self-dissimilarity between different classes of a complex system, allowing a model-free comparison [8]. However, it has also been suggested that much of this information is effectively encrypted, with the heritable information being encoded with random keys from the environment [9]. As such, these random keys are recorded as frozen accidents and increase the operative information content, as well as help direct the system during the process of evolution, producing objects that can construct other objects [10]. This is significant since one important characteristic of objects produced autonomously by machinery (such as life), which itself is instructed in some way, is their relative complexity as compared to objects that require no information for their assembly, beyond what chemistry and physics alone can provide. This means that for complex objects there is “object-assembly” information that is generated by an evolutionary system and is not just the product of laws of physics and chemistry alone. Biological systems are the only known source of agency in the universe [11], and it has been suggested that new physical laws are needed to understand the phenomenon of life [12]. The challenge is how to explore the complexity of objects generated by evolutionary systems without a priori having a model of the system.

Herein, we present the foundations of a new theoretical approach to agnostically bound the amount of information required to construct an object, via an “assembly” process. This is achieved by considering how the object can be deconstructed into its irreducible parts and then evaluating the minimum number of steps necessary to reconstruct the object along any pathway. The analysis of assembly is done by the recursive deconstruction of a given object using the shortest paths, and this can be used to evaluate the effective assembly index for that object [13]. In developing assembly theory, we have been motivated to create an intrinsic measure of an object forming through random processes, where the only knowledge required of the system is the basic building blocks and the permitted ways of joining structures together. This allows us to determine when an extrinsic agent or evolutionary system is necessary to construct the object, permitting the search for complexity in the abstract, without any specific notions of what we are looking for. Thus, we remove the requirement for an external imposition of meaning (see Figure 1).

The development of the assembly index [13] was motivated by the desire to define a biological threshold, such that any object found in abundance with an assembly index above the threshold would have required the intervention of one or more biological processes to form [16]. The assembly index of an object is the length of the shortest pathway to construct the object starting from its basic building blocks. It should be noted that this approach is entirely classical [17], allowing the quantification of pathways through assembly space probabilistically as a way to understand what life does. We construct the object using a sequence of joining operations, where at each step any structures already created are available for use in subsequent steps; see Figure 2. The shortest pathway approach is in some ways analogous to Kolmogorov complexity [15], which in the case of strings is the shortest computer program that can output a given string. However, assembly differs in that we only allow joining operations as defined in our model. This restriction is intended to allow the assembly process to mimic the natural construction of objects through random processes, and it also importantly allows the assembly index of an object to be computable for all finite objects (see Theorem 4 in Section 3.5). Importantly, the assembly index is measurable for molecules, which further sets assembly apart from Kolmogorov complexity and other measures of algorithmic information. The assembly process can also be compared to the concept of thermodynamic depth [18], which is defined as the amount of information that is needed to specify which of the possible trajectories a system followed to reach a given state.

Given a system where objects interact randomly and with equal probability, it is intuitively clear that the likelihood of an object being formed in *n* steps decreases rapidly with *n*. However, it is also true that a highly contrived set of biases could guarantee the formation of any object. For example, this could occur if we were to model the system such that any interactions contributing to the formation of the object were certain to be successful, while other interactions were prohibited. For complex objects, such a serendipitous set of biases would seem unlikely in the absence of external information about the end products, but physical systems generally do have biases in their interactions, and we can explore how these affect the likelihood of the formation of objects. However, we expect that for any perceived “construction processes” that require a large enough set of highly contrived biases, we can deduce that external information is required in the form of a “machine” that is doing the construction. In our recent work on molecular complexity, this notion was explored through the construction of a probabilistic model, in which steps through an assembly pathway were modelled as choices on a decision tree (see the supplementary information of [19]), with the probability of choices drawn from a random distribution. By then adding different levels of bias to that distribution, we explored the change in probability of the most probable pathway, as the path length increased. We found that even in the case of fairly substantial bias, the highest path probability drops significantly with path length. Therefore, an abundance of objects with high enough assembly indices would need specific sequences of biases that are beyond what one would expect an abiotic system that relies only on the information encoded by the laws of physics to provide. The location of that threshold will be system-dependent, but we can be confident a threshold region exists, above which objects in an assembly space require external processes to reach. The processes that allow for the crossing of that threshold may be critical to study to determine how life can happen.

Technological processes are bootstrapped to biological ones, and hence, by extension, the production of technosignatures involves processes that necessarily have a biological origin. Examples of biosignatures and technosignatures include chemical products produced by the action of complex molecular systems such as networks of enzymes [20] and objects whose creation involved any biological organisms such as technological artefacts [21], complex chemicals made in the laboratory [22], and the complete works of Shakespeare. Finding the object in some abundance, or a single object with a large number of complex, but precisely repeating features, is required in order to distinguish single random occurrences from deliberately generated objects. For example, a system which produces long random strings will generate some that have a high assembly index, but not in abundance. Finding the same long string more than once will tell us that there is a bias in the system towards creating that string; thus, searching for signatures of life should involve looking for objects with a high assembly index found in relatively high abundance. We can also deduce biological origin from repeated structures with a high assembly index, found within single objects—for example, repeated complex phrases within a long string. This approach would allow us to determine the biological origin of a Shakespeare play without knowing anything about language or grammar.

In this manuscript, we explore the foundations of assembly theory, as well as some of its properties and variants, and determine bounds on the assembly index. We offer some examples of the use of assembly theory in systems of varying dimensionality and describe some potential real-world applications of this approach.

## 2. Results

### 2.1. Graph Theoretical Prerequisites

In constructing an assembly space, we consider a set of objects, possibly infinitely many objects, which can be combined in various ways to produce others. If an object a can be combined with some other object to yield an object b, we represent the relationship between a and b by drawing a directed edge or arrow from a to b. Altogether, this structure is a quiver, also called a directed multigraph, as we allow for the possibility that there is more than one way to produce b from a; that is, there may be more than one edge from a to b.

**Definition** **1.***A **quiver*** Γ *consists of*
1.*A set of vertices $V\left(\Gamma \right)$;*2.*A set of edges $E\left(\Gamma \right)$;*3.*A pair of maps $s_{\Gamma },t_{\Gamma }:E\left(\Gamma \right)\to V\left(\Gamma \right)$.*

For an edge $e\in E\left(\Gamma \right)$, $s_{\Gamma }\left(e\right)$ is referred to as the source and $t_{\Gamma }\left(e\right)$ the target of the edge, and we will often leave off the subscripts when the context is clear, e.g., s and t. We will often describe an edge $e\in E\left(\Gamma \right)$ with $s\left(e\right)=a$ and $t\left(e\right)=b$ as $e∼\left[ba\right]$. This does not mean that *e* is a unique edge with endpoints *a* and *b*; it is possible that two edges e≠f have the same endpoints $e∼f∼\left[ba\right]$.

From here, we consider paths—that is, sequences of edges—that describe the process of sequentially combining objects to yield intermediate objects and ultimately some terminal object.

**Definition** **2.***If* Γ*is a quiver, a **path*** $\gamma =a_{n}\ldots a_{1}$*in* Γ*of length*n≥1*is a sequence of edges, such that* $t\left(a_{i}\right)=s\left(a_{i+1}\right)$*for*1≤i≤n−1*. The functions* s*and* t*can be extended to paths as*$s\left(\gamma \right)=s\left(a_{1}\right)$*and* $t\left(\gamma \right)=t\left(a_{n}\right)$*. We write* $\left|\gamma \right|$* to denote the length, or number of edges, in the path. Additionally, for each vertex* x∈Γ*there is a **zero path**, denoted* $e_{x}$*, with length 0 and*$s\left(e_{x}\right)=t\left(e_{x}\right)=x$.

A natural point is that combining two objects should never yield something that can be used to create either of those objects. Essentially, there are no directed cycles—sequences of edges that form a closed cycle—within the quiver.

**Definition** **3.***A path* γ*in a quiver* Γ*is a **directed cycle** if* $\left|\gamma \right|\geq 1$*with* $t\left(\gamma \right)=s\left(\gamma \right)$.

**Definition** **4.***A quiver*Γ*is **acyclic** if it has no directed cycles*.We can think of an object b as being *reachable* from an object a if there is a path from a to b, and this relationship forms a partial ordering on the quiver if the quiver is acyclic.

**Definition** **5.***Let* Γ*be an acyclic quiver and let* $x,y\in V\left(\Gamma \right)$*. We say*y*is **reachable** from* x*if there exists a path* γ*such that* $s\left(\gamma \right)=x$*and*$t\left(\gamma \right)=y$*, where* $\left|\gamma \right|\geq 0$.

**Lemma** **1.***Let*Γ*be an acyclic quiver, and define a binary relation*≤*on the vertices of* Γ*such that* x≤y*if and only if* y*is reachable from* x. $\left(V\left(\Gamma \right),\leq \right)$*is a partially ordered set, and* ≤*is referred to as the **reachability relation** on* Γ.

**Proof.** For ≤ to be a partial ordering on $V\left(\Gamma \right)$, we need to show that it is reflexive, transitive and antisymmetric. Reflexivity follows directly from the definition of reachability as x is reachable from itself via the zero path $e_{x}$. To show transitivity, let a≤b and b≤c. If a=b or b=c, then we are done. Otherwise, there are paths $\gamma_{ba}=u_{m}\ldots u_{1}$ from a to b and $\gamma_{cb}=v_{n}\ldots v_{1}$ from b to c. The composite path $\gamma_{cb}\circ \gamma_{ba}=v_{n}\ldots v_{1}u_{n}\ldots u_{1}$ is a path from a to c; thus c is reachable from a so that a≤c. Now consider antisymmetry and suppose that a≤b and b≤a. Then there exist paths $\gamma_{ba}$ and $\gamma_{ab}$ from *a* to *b* and b to a, respectively. Then $\gamma_{ab}\circ \gamma_{ba}$ is a path from a to itself. Since Γ is acyclic, this implies that $\gamma_{ab}\circ \gamma_{ba}=e_{a}$, and consequently that $\gamma_{ab}=\gamma_{ba}=e_{a}$. Thus, a=b and ≤ is antisymmetric. □

The idea of reachability allows us to think of all objects that are reachable from (or above) a given object x, the upper quiver of x. Similarly, we can think of all objects that can reach x, the lower quiver.

**Definition** **6.***Let*Γ*be an acyclic quiver and let*≤*be the reachability relation on it. The **upper quiver** of* $x\in V\left(\Gamma \right)$*is* x↑*with vertices*$V\left(x↑\right)=\{y\in V\left(\Gamma \right)|x\leq y\}$*, edges*$E\left(x↑\right)=\left\{e\in E\left(\Gamma \right)|s_{\Gamma }\left(e\right),t_{\Gamma }\left(e\right)\in V\left(x↑\right)\right\}$*,*$s_{x↑}=s_{\Gamma }{|}_{E\left(x↑\right)}$*, and*$t_{x↑}=t_{\Gamma }{|}_{E\left(x↑\right)}$*. The **lower quiver** of*$x\in V\left(\Gamma \right)$*is*x↓*with vertices* $V\left(x↓\right)=\{y\in V\left(\Gamma \right)|y\leq x\}$*, edges*$E\left(x↓\right)=\left\{e\in E\left(\Gamma \right)\;|s_{\Gamma }\left(e\right),t_{\Gamma }\left(e\right)\in V\left(x↓\right)\right\}$*,*$s_{x↓}=s_{\Gamma }{|}_{E\left(x↓\right)}$*, and* $t_{x↓}=t_{\Gamma }{|}_{E\left(x↓\right)}$.

Similarly, the upper quiver of a subset $Q\subseteq V\left(\Gamma \right)$ in Γ is Q↑ with vertices $V\left(Q↑\right)=\{y\in V\left(\Gamma \right)|\left(\exists q\in Q\right)q\leq y\}$, edges $E\left(Q↑\right)=\{e\in E\left(\Gamma \right)|s_{\Gamma }\left(e\right),t_{\Gamma }\left(e\right)\in V\left(Q↑\right)\}$, $s_{Q↑}=s_{\Gamma }{|}_{E\left(Q↑\right)}$, and $t_{Q↑}=t_{\Gamma }{|}_{E\left(Q↑\right)}$. The lower quiver of a subset is defined dually.

Going further, we can consider those objects that cannot be reached as *minimal* and those that cannot reach anything as *maximal*. An object which can be reached by finitely many objects is called *finite.*

**Definition** **7.***Let*Γ*be an acyclic quiver,*≤*be the reachability relation on it, and* x*a vertex in* Γ*. Then,* x*is said to be **maximal** in* Γ *if, whenever*x≤y*in* Γ*, we have* x=y*. Dually,* x*is **maximal** in* Γ*if, whenever* y≤x*in* Γ*, we have* x=y*. The set of all maximal vertices of*Γ* is denoted* $max\left(\Gamma \right)$*with* $min\left(\Gamma \right)$*defined dually.*

**Definition** **8.***A quiver*Γ*is said to be **finite****if its vertex and edge sets are both finite. Similarly, a vertex* x*in a quiver* Γ*is said to be finite if*x↓*in* Γ*is a finite quiver.*

With this idea of a quiver of objects defined, we can consider asking about subsets of objects and relations between them in the context of the quiver as a whole.

**Definition** **9.***Let*Γ*and* $\Gamma {}^{\prime }$*be quivers. Then* $\Gamma {}^{\prime }$*is a **subquiver** of* Γ*if*$V\left(\Gamma {}^{\prime }\right)\subseteq V\left(\Gamma \right)$*,*$E\left(\Gamma {}^{\prime }\right)\subseteq E\left(\Gamma \right)$*,*$s_{\Gamma {}^{\prime }}=s_{\Gamma }{|}_{E\left(\Gamma {}^{\prime }\right)}$ *and*$t_{\Gamma {}^{\prime }}=t_{\Gamma }{|}_{E\left(\Gamma {}^{\prime }\right)}$*. We will denote this relationship as* $\Gamma {}^{\prime }\subseteq \Gamma$.

**Lemma** **2.***If*X*,*Y*, and* Z*are quivers, such that* X⊆Y*and* Y⊆Z*, then* X⊆Z*. That is, the binary relation* ⊆*on quivers is transitive.*

**Proof.** Suppose X, Y, and Z are quivers with X⊆Y and Y⊆Z. Then, $V\left(X\right)\subseteq V\left(Y\right)\subseteq V\left(Z\right)$, so that $V\left(X\right)\subseteq V\left(Z\right)$. Similarly, $E\left(X\right)\subseteq E\left(Z\right)$. Next, since $s_{X}=s_{Y}{|}_{E\left(X\right)}$, $s_{Y}=s_{Z}{|}_{E\left(Y\right)}$ and $E\left(X\right)\subseteq E\left(Y\right)$, $s_{X}=s_{Z}{|}_{E\left(X\right)}$. The same argument applies to show that $t_{X}=t_{Z}{|}_{E\left(X\right)}$. Thus X⊆Z, so that ⊆ is transitive. □

Finally, we will need to consider how to map one quiver to another in a consistent fashion, maintaining the basic relational structure of the original quiver.

**Definition** **10.***Let*Γ*and* $\Gamma {}^{\prime }$*be quivers. A **quiver morphism**, denoted*$m:\Gamma \to \Gamma {}^{\prime }$*, consists of a pair* $m=\left(m_{v},m_{e}\right)$*of functions*$m_{v}:V\left(\Gamma \right)\to V\left(\Gamma {}^{\prime }\right)$*and* $m_{e}:E\left(\Gamma \right)\to E\left(\Gamma {}^{\prime }\right)$*such that* $m_{v}\circ s_{\Gamma }=s_{\Gamma {}^{\prime }}\circ m_{e}$*and* $m_{v}\circ t_{\Gamma }=t_{\Gamma {}^{\prime }}\circ m_{e}$*. That is, the following diagrams commute:*

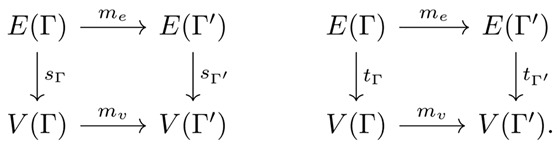



### 2.2. Assembly Spaces

The assembly process is the process of constructing some object, which can be decomposed into a finite set of basic objects, through a sequence of joining operations. During this process, objects already constructed can be used in subsequent steps. We formally define this in the context of an assembly space (see Figure 3), as follows:

**Definition** **11.**
*An **assembly space** is an acyclic quiver*

Γ

*together with an edge-labelling map*

$\phi :E\left(\Gamma \right)\to V\left(\Gamma \right)$

*which satisfies the following axioms:*
1.

$min\left(\Gamma \right)$

*is finite and non-empty;*
2.

$\Gamma =min\left(\Gamma \right)↑$

*;*
3.If a*is an edge from* x*to*z*in* Γ*with* $\phi \left(a\right)=y$*, then there exists an edge* b*from* y*to* z*with* $\phi \left(b\right)=x$.


**Definition** **12.***The set of minimal vertices of an assembly space*Γ*is referred to as the **basis** of* Γ*and is denoted* $B_{\Gamma }$*. Elements of the basis are referred to as basic objects, basic vertices, or basic elements.*

An assembly space as in definition 11 is denoted $\left(\Gamma ,\;\phi \right)$, or simply Γ where appropriate. x∈Γ is taken to mean that x is a vertex of the quiver. Within the assembly space, we can think of the vertices as objects and traversal along the directed edge as the construction of the target object from the source object, with the edge label determining the object that is combined with the source to construct the target. The assembly process starts from a set of basic objects (axiom 1) from which all other objects can be constructed (axiom 2). Axiom 3 requires that a symmetric edge exists for every edge within the assembly space wherein the roles of source and edge label are reversed. Intuitively, this can be thought of as saying: if you can combine x with y to construct z, then you can also combine y with x to construct z. Axiom 3 also formalises the requirement that both items in the construction lie below the target in the assembly tree, i.e., only objects already assembled can be used in further assembly steps (see Lemma 3).

**Lemma** **3.**
*Let $\left(\Gamma ,\phi \right)$ be an assembly space and let x∈Γ. If $e∼\left[ba\right]$ is an edge in Γ with a,b∈x↓, then $\phi \left(e\right)\in x↓$.*


**Proof.** Since Γ is an assembly space, we have $\phi \left(e\right)\leq b$ where ≤ is the reachability relation on Γ, since there is an arrow from $\phi \left(e\right)$ to b by point 3 of Definition 11. By construction, b≤x so that $\phi \left(e\right)\leq x$. Therefore, $\phi \left(e\right)\in x↓$. □

Within an assembly space, an assembly pathway is a sequence that respects the order of the reachability relation. We can think of an assembly pathway as being an order of construction for all the objects within the space, ensuring that the objects required for each step are available earlier in the sequence.

**Definition** **13.***An **assembly pathway** of an assembly space*Γ*is any topological ordering of the vertices of* Γ*with respect to the reachability relation.*

**Definition** **14.***An assembly space*Γ*with reachability relation* ≤*is said to be **split-branched** if for all* x,y∈Γ*,*x≤y*or*y≤x*whenever* $V\left(x↓\right)\cap V\left(y↓\right)\neq \emptyset$.

In a split-branch assembly space, other than basic objects, when combining two different objects, neither of them can have an assembly pathway that uses objects created in the construction of the other. They may use objects that are considered identical (e.g., the same string) but these are separate objects within the space. Since we can define an assembly map to a new space where these separate but identical objects are mapped to the same object, the split-branched assembly index for a system is an upper bound for the assembly index on that system (see Section 3.4). Calculations of the assembly index in a split-branch space can be less computationally intensive than in the corresponding non-split-branch space. A split-branch algorithm was used in our recent work on molecular assembly [19].

### 2.3. Assembly Subspaces and the Assembly Index

We define an assembly subspace, and the rooted property, as follows:

**Definition** **15.***Let* $\left(\Gamma ,\;\phi \right)$*and*$\left(\Gamma {}^{\prime },\psi \right)$*be assembly spaces. Then*$\left(\Gamma {}^{\prime },\psi \right)$*is an****assembly*** ***subspace****of*$\left(\Gamma ,\;\phi \right)$*if*$\Gamma {}^{\prime }$*is a subquiver of*Γ*and*$\psi =\phi {\left.\right|}_{E\left(\Gamma {}^{\prime }\right)}$*. This relationship is denoted as*$\left(\Gamma {}^{\prime },\psi \right)\subseteq \left(\Gamma ,\;\phi \right)$*, or simply*$\Gamma {}^{\prime }\subseteq \Gamma$*, when there is no ambiguity.*

**Definition** **16.**
*Let*

$\Gamma {}^{\prime }$

*be an assembly subspace of*

Γ

*. Then*

$\Gamma {}^{\prime }$

*is*
**
*rooted*
**
*in*

Γ

*if*

$B_{\Gamma }$

*is non-empty, and*

$B_{\Gamma^{\prime }}\subseteq B_{\Gamma }$

*as sets.*


An assembly subspace of Γ is simply an assembly space that contains a subset of the objects in Γ and the relationships between them. It is rooted if its set of basic objects is a nonempty subset of the basic objects of Γ. The assembly subspace relationship is transitive (see Lemma 4).

**Lemma** **4.**
*Let U, V, and W be assembly spaces with U⊆V and V⊆W, then U⊆W. Further, if U is rooted in V and V is rooted in W, then U is rooted in W.*


**Proof.** Let $\left(U,\;\phi_{U}\right),\;\left(V,\;\phi_{V}\right)$, and $\left(W,\;\phi_{W}\right)$ be assembly spaces such that $\left(U,\;\phi_{U}\right)\subseteq \left(V,\;\phi_{V}\right)$ and $\left(V,\;\phi_{V}\right)\subseteq \left(W,\;\phi_{W}\right)$. Since U, V and W are quivers, U⊆W by the transitivity of ⊆ on quivers. Further, since $\phi_{U}=\phi_{V}{\left.\right|}_{E\left(U\right)}$, $\phi_{V}=\phi_{W}{\left.\right|}_{E\left(V\right)}$, and $E\left(U\right)\subseteq E\left(W\right)$, we have $\phi_{U}=\phi_{W}{\left.\right|}_{E\left(W\right)}$. Thus, $\left(U,\;\phi_{U}\right)\subseteq \left(W,\;\phi_{W}\right)$. That is, ⊆ is transitive on assembly spaces. If U is rooted in V and V is rooted in W, then $B_{U}\subseteq B_{v}\subseteq B_{W}$. That is, U is rooted in W. □

We can also show that for any object x in an assembly space Γ, the objects and relationships that lie below x are a rooted assembly subspace of Γ.

**Lemma** **5.***Let*$\left(\Gamma ,\phi \right)$*be an assembly space and let*x∈Γ*. Then,*$\left(x↓,\phi {|}_{x↓}\right)$*is a rooted assembly subspace of*Γ.

**Proof.** We first show that $\left(x↓,\phi {|}_{x↓}\right)$ is an assembly space. Since $\left(\Gamma ,\phi \right)$ is an assembly space, it is the upper set of its basis, $B_{\Gamma }$. As such, $min\left(x↓\right)$ is a non-empty subset of $B_{\Gamma }$ and $x↓=\min\left(x↓\right)↑$, giving us axiom 1. The remaining axiom follows directly from Lemma 6. Additionally, we already have that $min\left(x↓\right)=B_{x↓}\subseteq B_{\Gamma }$, so x↓ is rooted in Γ. □

**Figure 3 entropy-24-00884-f003:**
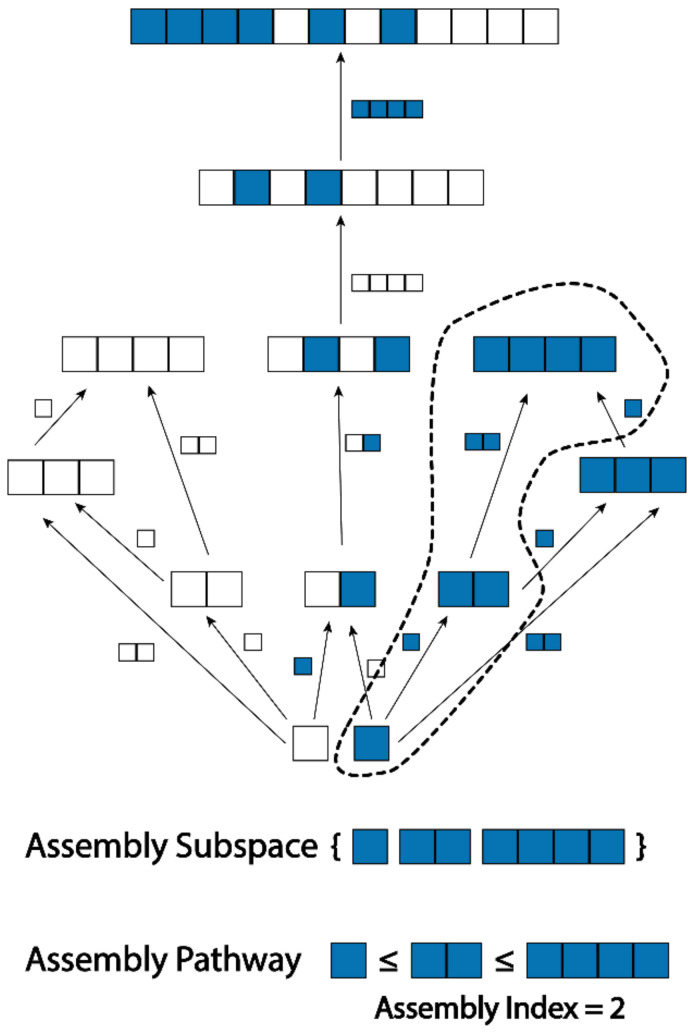
An assembly space comprised of objects formed by joining together white and blue blocks. Some of the arrows have been omitted for clarity. The dotted region is an assembly subspace, and the topological ordering of the objects in the subspace represents a minimal assembly pathway for any subspace containing the sequence of four blue boxes.

We now move on to the assembly index, which is a measure of how directly an object can be constructed from basic objects.

**Definition** **17.***The **cardinality** of an assembly space*$\left(\Gamma ,\phi \right)$*is the cardinality of the underlying quiver’s vertex set,*$\left|V\left(\Gamma \right)\right|$*. The augmented cardinality of an assembly space*$\left(\Gamma ,\phi \right)$*with basis*$B_{\Gamma }$*is*$\left|V\left(\Gamma \right)\backslash B_{\Gamma }\left|=\left|V\left(\Gamma \right)\right|-\right|B_{\Gamma }\right|$.

**Definition** **18.***The **assembly index*** $c_{\Gamma }\left(x\right)$*of a finite object*x∈Γ*is the minimal augmented cardinality of all rooted assembly subspaces containing*x*. This can be written*$c\left(x\right)$*when the relevant assembly space*Γ*is clear from the context.*

The cardinality is the number of objects within the assembly space, and the augmented cardinality is the number of objects excluding basic objects. Thus, the assembly index of x is the number of objects within the smallest rooted assembly subspace containing x, not including the basic objects. We require the subspaces to be rooted, as otherwise, a space containing only x would fit this criterion.

The assembly index can be thought of as how many construction steps we need to take at a minimum to create x, starting from our set of basic objects. This is a key concept in assembly theory, as it allows us to place a lower bound on the number of joining operations required to make an object. The augmented cardinality is used as defining the assembly index without including basic objects in accord with this physical interpretation of joining objects in steps; however, the cardinality could instead be used if desired, and the difference in the measures for any structures with shared basic objects would be a constant.

### 2.4. Assembly Maps

An assembly map is defined as follows:

**Definition** **19.**
*Let*

$\left(\Gamma ,\phi \right)$

*and*

$\left(\Delta ,\psi \right)$

*be assembly spaces. An **assembly map** is a quiver morphism*

f:Γ→Δ

*such that*

$\psi \circ f_{e}=f_{v}\circ \phi$

*. That is, the following diagram commutes:*


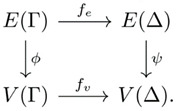



An assembly map (see Figure 4) is essentially a mapping from one assembly space Γ to another Δ, which maintains relationships between objects but may map multiple objects in Γ to the same object in Δ. One such map that is generally applicable is the mapping of an assembly space to the space of integers under addition, in which each object maps to an integer representing the number of basic objects it is comprised of.

Assembly maps can be useful for finding a lower bound to the assembly index (see Section 3.4), which can allow for mapping to systems that may be more computationally tractable than the main system of interest. The following theorem provides a basis for the lower bounds, the essential point being that the image of an assembly space under an assembly map is an assembly space.

**Theorem** **1.**
*If f:Γ→Δ is an assembly map between assembly spaces $\left(\Gamma ,\phi \right)$ and $\left(\Delta ,\psi \right)$, then $\left(f\left(\Gamma \right),\varphi \right)$ with $\varphi =\psi {|}_{E\left(f\left(\Gamma \right)\right)}$ as an assembly subspace of Δ.*


**Proof.** See Appendix B. □

### 2.5. Bounds on the Assembly Index

In this section, we look at some bounds on the assembly index. First, the assembly index of an object x in an assembly space Γ is always less than or equal to the assembly index of x in any rooted assembly subspace of Γ that contains x. Essentially, since the assembly subspace may have fewer edges, and cannot have more edges, there are fewer “shortcuts” for assembling a given object.

**Lemma** **6.***Let*X*be an assembly space and*Y*a rooted assembly subspace of*X*. For every finite*y∈Y*, the assembly index of*y*in*Y*is greater than or equal to the assembly index of*y*in*X*. That is,*$c_{Y}\left(y\right)\geq c_{X}\left(y\right)$*for all*y∈Y.

**Proof** .Let y∈Y and suppose $c_{Y}\left(y\right)<c_{X}\left(y\right)$. Then, there exists a rooted assembly subspace Z⊆Y containing y, such that $\left|Z\backslash B_{Z}\right|=c_{Y}\left(y\right)$. However, by the transitivity of rooted assembly subspaces (Lemma 4), Z is a rooted assembly subspace of X —but if that is the case, there exists a rooted assembly subspace of X with augmented cardinality less than $c_{X}\left(y\right)$, namely Z; a contradiction. □

Since the lower quiver of an object x is a rooted assembly subspace, we know the assembly index of the object in x↓ bounds the real assembly index of the object from above. However, we can show that these assembly indices are equal, i.e., $c_{\Gamma \;}\left(x\right)=c_{x↓}\left(x\right)$. This result allows any computational approaches aiming to compute $c\left(x\right)$ to focus only on the objects below x.

**Theorem** **2.**
*Let Γ be an assembly space and let x∈Γ be finite. Then $c_{\Gamma }\left(x\right)=c_{x↓}\left(x\right)$.*


**Proof.** Since x↓ is finite, we need only consider finite, rooted assembly subspaces of Γ. Let Δ⊆Γ be such a subspace containing x, and suppose that Δ⊈x↓. Let y∈Δ such that y∉x↓, then $\left(\Delta \backslash y↑\right)$ is a rooted assembly subspace of Γ containing x with augmented cardinality strictly less than Δ. As such $\left|\Delta \backslash B_{\Delta }\right|\neq c_{\Gamma }\left(x\right)$. □

In other words, if Δ is not a subspace of x↓, then it cannot have the augmented cardinality $c_{\Gamma }\left(x\right)$. Thus, by contrapositive if $\left|\Delta \backslash B_{\Delta }\right|=c_{\Gamma }\left(x\right)$, then Δ⊆x↓. Since Δ is rooted in Γ, it must also be rooted in x↓.

Therefore, if a rooted subspace of Γ has the minimal augmented cardinality in Γ, it must be a rooted assembly subspace of x↓. This implies that $c_{\Gamma }\left(x\right)\geq c_{x↓}\left(x\right)$. Additionally, by Lemma 6, $c_{\Gamma }\left(x\right)\leq c_{x↓}\left(x\right)$. Then, $c_{\Gamma }\left(x\right)=c_{x↓}\left(x\right)$. □

Finally, assembly maps allow us to place lower bounds on the assembly index – the assembly index of the image of an object bounds the object’s actual assembly index below. In other words, we can place lower bounds on the assembly index of an object by mapping the assembly space into a simpler space and computing the assembly index there.

**Theorem** **3.***If*f:Γ→Δ*is an assembly map, then*$c_{f\left(\Gamma \right)}\left(f\left(x\right)\right)\leq c_{\Gamma }\left(x\right)$*for all finite *x∈Γ.

**Proof.** See Appendix B. □

### 2.6. Computability

The determination of the assembly index can be computationally challenging for more complex objects. However, importantly, the assembly index is computable, as shown below.

**Theorem** **4.***If*Γ*is an assembly space and*x∈Γ*is finite, with*x↓*finite, then*$c\left(x\right)$*is computable*.

**Proof.** As shown in the proof of theorem 2, every rooted assembly subspace with minimal augmented cardinality and containing x is a minimal rooted assembly subspace of x↓. Since x↓ is finite, the set of assembly subspace of x↓ is finite, and each such subspace is finite. Consequently, the basis of each subspace is computable. As such, the set of all rooted subspaces is computable. The cardinality of each subspace is computable, so the set of cardinalities of all rooted subspaces is computable. Finally, the minimum of a finite set of natural numbers is computable. Therefore, $c_{\Gamma }\left(x\right)$ is computable. □

Example algorithms for determining the assembly index can be found in Appendix B.

## 3. Discussion

The application of assembly spaces allows us to consider paths through the space of possible objects that could be created through a model of joining through random interactions. We can find the shortest possible path to create an object, in the context of other possible objects that could have been created at each step, and come to a judgement on whether a random system (even a biased one) could have selected for a population of that object without additional information. Additional information here refers to more than is contained within our system of starting objects and joining rules.

The assembly approach does not perfectly model the workings of physical systems. It is a necessarily simplified model. For example, in our recent work on molecules [19], we assumed that any structure could be created by joining two others, not considering chemical feasibility (other than following valence rules) or modelling steps where molecules partly fall apart. What we can do, however, is use the assembly process as a structural complexity model in our much simpler and generally more permissive system. We can say, for a particular molecule, that even if we were to discard the regular restrictions of synthetic chemistry, it would be impossible to make this molecule in fewer than *n* steps, and then judge how much more difficult it would be if we were to reinstate those restrictions. For example, if we were to amend the molecular model by removing unrealistic chemicals from our assembly space, the resulting space would be an assembly subspace with an assembly index equal to or higher than the original (see Lemma 6 in Section 2.5). Still, that model would not be a perfect synthetic assembly space, but it highlights the principle that adding more restrictions tends to make assembly more difficult.

The assembly index can be used to determine a rough threshold above which the biases required to create an object are beyond what can be accounted for by the model. Objects significantly above that threshold can be considered biosignatures. It is important to stress that we are not arguing that the threshold cannot be crossed without biology, since a biological system developing from purely abiotic ones has clearly happened at least once. However, this would require systems outside of our model, such as replication, duplication, and evolution. The exploration of the objects and processes that cross this threshold will be of key importance to our understanding of life.

In the following sections, we discuss how the formalisation of assembly spaces could be used to explore a variety of systems of varying dimensionality; see Figure 5.

### 3.1. Addition Chains

One of the simplest assembly spaces is the space of positive integers under addition. This is a space Γ, ϕ where VΓ=ℕ\0=1,2,3,…, the set of positive integers, and for each edge e~zx, if ϕe=y, then x+y=z. In other words, in traversing the assembly space along an edge, the target vertex is the source vertex plus the edge label.

An assembly pathway within some finite rooted assembly subspace of Γ is equivalent to an addition chain, which is defined [23] as “a finite sequence of positive integers $1=a_{0}\leq a_{1}\ldots \leq a_{r}=n$ with the property that for all i>0 there exists j,k with $a_{i}=a_{j}+a_{k}$ and r ≥ i>j ≥ k ≥ 0”. In other words, an addition chain is a sequence of integers, starting with 1, in which each integer is the sum of two integers (not necessarily unique) that appear previously in the sequence. A minimal, or optimal, addition chain for an integer is an addition chain of the shortest possible length terminating in that integer. An example of an optimal addition chain for n=123 is
$$\left\{1,\;2,\;3,\;5,\;10,\;15,\;30,\;60,\;63,\;123\right\}$$

The length of the minimal addition chain (subtracting 1 to account for the single basic object) is equivalent to the assembly index of that integer in Γ. Assembly spaces can generally be mapped to the space of integers through an assembly map by mapping each object to an integer representing its size (i.e., the number of basic objects contained within it), see Figure 4. This allows us to determine a lower bound for the assembly index for complex spaces, although the lower bound may be substantially lower than the actual assembly index, in which case, other assembly maps may be more suitable.

### 3.2. Vectorial Addition Chains

Addition chains can be further generalised to vectorial addition chains [24]. We define a vectorial addition chain for a k-dimensional vector of natural numbers n∈ℕk/0 (excluding the zero vector) as a sequence of ai∈ℕk/0 such that for −k+1≤i≤0, ai are the standard basis of unit vectors 1, 0, …,0, 0, 1, …, 0, …, 0, 0, …, 1 and for each i>0  there exists a j, k with ai=aj+ak and i>j≥k. An example of a vectorial addition chain for 8,8,10 is
1,0,0,0,1,0,0,0,1,1,1,0,1,1,1,2,2,2,4,4,4,8,8,8,8,8,9,8,8,10

As with the addition chain assembly space Γ, we can define an assembly space Δ based on vectorial addition chains. An assembly map exists from Δ to Γ, involving summing each of the vectors, and thus the assembly index in Γ is a lower bound for the assembly index in Δ by Theorem 3. Δ can also provide a useful lower bound to other assembly spaces, which have basis B, such that $\left|B\right|>1$, where the vectors comprising the vertices of Δ represent a count of each of the different basic objects within the corresponding vertex of Γ. For example, in the case of shapes constructed from red and blue blocks, all shapes made of 3 red and 4 blue blocks would map to the vector $\left\{3,\;4\right\}$.

### 3.3. Strings

In one-dimensional strings, we can define an assembly space $\left(\Gamma ,\phi \right)$ of strings, where each $s\in V\left(\Gamma \right)$ is a string and if a string z can be produced by concatenating strings x and y, then there exists an edge $e\sim \left[zx\;\right]$ with $\phi \left(e\right)=y$ (see Figure 6). There are multiple systems that have string representations, including text strings, binary signals and polymers.

Alternative methods for analysing the complexity/information content of strings are the Shannon information [14] and Kolmogorov complexity [15]. For a string S that can be in N possible states $\left\{s_{1}\ldots s_{N}\right\},$ according to an observer, the Shannon entropy is a measure of the uncertainty of which state S is in according to the observer. If the states of the string have probabilities $\left\{p_{1}\ldots p_{N}\right\}$, then the Shannon entropy of S is given by $$H\left(S\right)=-\overset{N}{\underset{i=1}{\sum }}p_{i}\log p_{i}$$ where a suitable base of the logarithm is selected depending on the desired units (e.g., base 2 for bits). Shannon information is the reduction in entropy on being provided with additional information about the probability distribution of the possible states. Entropy is maximum when all states are equally likely (p=1/N and $H\left(S\right)=\log N)$ and has a minimum $H\left(S\right)=0$ when $p_{i}=1$ for an *i*, i.e., the state of *S* is known.

The Kolmogorov complexity [15] of an object is the length of the shortest program that outputs that object, in a given Turing-complete language. Although Kolmogorov Complexity is dependent on the language used, it can be shown that the Kolmogorov complexity *C* in any language *ϕ* can be related to the Kolmogorov complexity in a universal language *U* by $C_{u}\left(c\right)\leq C_{\phi \left(x\right)}+c$ for a constant *c* [15]. If a string cannot be expressed in a universal language by a program shorter than its length, it is considered random. It has been shown that the Kolmogorov complexity is not computable, whereas the assembly index is computable (see Theorem 4).

### 3.4. Pixels and Voxels

We can extend the assembly process to two dimensions by considering a grid of pixels, or coloured boxes, for example, a digital image. For simplicity, we will consider images with black and white basic objects, although this could be simply extended to greyscale images or colour images (e.g., greyscale images could have 256 basic objects representing different pixel intensities, as in an 8-bit greyscale image). We can define an assembly space with assemblages of black and white pixels as objects. In this space, two assemblages a and x are connected by an edge $e\;\sim \;\left[xa\right]$ if a is a substructure of x. The edge *e* is labelled as (*e*) = *b* with *b* the complement of *a* in *x*. In other words, you can connect *a* and *b* together to get *x*. A choice can be made about whether to enforce the preservation of orientation, or whether to consider substructures rotated by 90 degrees to be equivalent, and the latter choice can be related to the former by way of an assembly map. An illustration of an assembly pathway in this space can be seen in Figure 7.

The assembly index can be mapped to the space of addition chains as normal and to a reduced representation of the image such as those generated by pooling operations used in convolutional neural networks, or quantisation matrices used in jpeg compression.

To extend assembly to three dimensions, we can consider structures created out of cubic building blocks, or voxels, as a natural extension of the two-dimensional model. Assembly theory does not need to be applied to objects as a whole, but can be applied to shared motifs or networks found within the objects [13], which can in some cases map to the problem of cubic building blocks. Assembly theory as described here currently has no simple extension to continuous objects; however, we can use an assembly map to define a function that consistently maps similar features to larger block structures, and can calculate the assembly index of that structural motif to explore whether it is over the biological threshold, if found in some abundance.

### 3.5. Graphs

An undirected graph $G\left(V,E\right)$ is defined by a set of vertices V and a set of edges E⊆V×V. An assembly space for connected graphs (directed or undirected) can be defined where Γ is the space of all connected graphs, with the basis set B consisting of a single node. The reachability relationship ≤ is defined on Γ such that $\phi \left(\left[G_{x},G_{a}\right]\right)=G_{b}$ if $V_{X}=V_{a}\cup V_{b}$ and $E_{x}=E_{a}\cup E_{b}\cup E_{ab}$ where $E_{ab}\;\subseteq V_{a}\times V_{b}$ and $E_{ab}\;\neq \emptyset \;$. In other words, $G_{x}$ contains all vertices and edges of $G_{a}$ and $G_{b}$ and also at least one edge between them. Similar spaces can be defined for graphs that are not necessarily connected by removing the requirement that $E_{ab}\;\neq \emptyset$. Vertex colours can be incorporated by expanding the basis set B. A graph assembly space can also be defined with edges as the basic objects, instead of vertices. Additional constraints allow for the study of spaces of other useful graph structures—for example, the restriction of vertex degree allows for the study of the space of molecular graphs [19]. As in the block structures, the assembly space of graphs can be used to analyse objects that have identical network motifs in them while not being identical in other ways. Assembly maps can be defined from the space of graphs to the space of addition chains, as a count of the number of vertices, and also to vectorial addition chains if the vertices are coloured.

### 3.6. Other Applications

There are various other examples where the assembly approach could be used to provide a useful analysis of objects. One example is in audio/electromagnetic signals—music. By utilising notes and silences as basic objects, possibly incorporating frequency/pitch, we could use assembly theory to distinguish natural signals such as those from a pulsar, or the sound of wind moving through a complex landscape, from sounds such as birdsong or structured communications. In such a system, abundance could be the same signal from multiple locations or from the same location but repeated. We can also consider the morphology of apparent geological formations to look for evidence of biological influence in the form of duplicated complex patterns.

Assembly theory can also be used to define a compression algorithm, such as the widely known Lempel–Ziv–Welch (LZW) algorithm [25]. In the LZW algorithm, repeated portions of text are represented by additional symbols in an expanded character set, and the need for a separate dictionary is removed by building the dictionary in such a way that it can be reconstructed during decompression. In an assembly-based implementation, we could initially calculate an assembly pathway for the string and then use the additional character set to indicate points at which substrings are duplicated or stored for re-use. It is unlikely that such a compression algorithm would be commercially useful due to the computational complexity of finding a minimal assembly pathway, but analysing compressibility in this way could provide further insights regarding the information content of string-like objects from an assembly space perspective.

## 4. Conclusions

Assembly theory can be used to explore the possible ways an object could have formed from its building blocks through random interactions, and we have now built on our prior work [13,19] by establishing a robust mathematical formalism. Through this, we can define a threshold above which extrinsic information from a biological source would have been required to create an observable abundance of an object because it is too improbable to have formed in abundance otherwise. The assembly index of an object, when above the threshold, can be used as an agnostic biosignature, giving a clear indication of the influence of information in constructing objects (e.g., via biological processes) without knowledge of the system that produced the end product. In other words, it can be used to detect biological influence even when we do not know what we are looking for [19]. Of interest is the ability to search for new types of life forms in the lab, alien life on other worlds, as well as identifying the conditions under which the random world embarks on the path towards life, as characterised by the emergence of physical systems that produce objects with a high assembly index. As such, assembly theory might enable us to not only look for the abiotic-to-living transition, identifying the emergence of life, but also to identify technosignatures associated with intelligent life with even higher assembly indices within a unified quantitative framework. We, therefore, feel that the concepts of assembly theory can be used to help us explore the universe for structures that must have been produced using an information-driven construction process; in fact, we could go as far as to suggest that any such process requiring information is a biological or technological process. This also means that assembly theory provides a new window on the problem of understanding the physics of life simply because the physics of information is the physics of life. We believe that such an approach might help us reframe the question from the philosophy of what life is [26] to a physics of what life does.

## Figures and Tables

**Figure 1 entropy-24-00884-f001:**
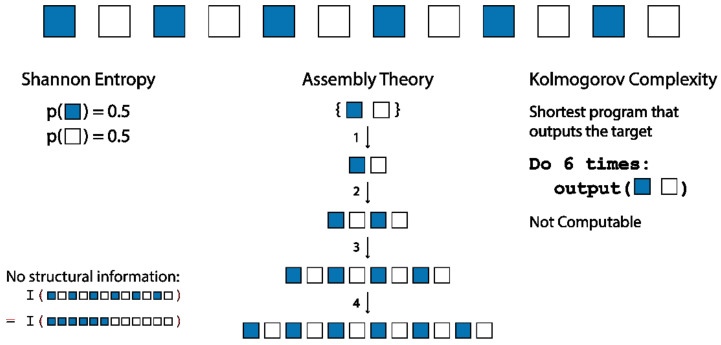
The assembly process (centre) [13] is compared to the implementations of Shannon entropy [14] (left) and Kolmogorov complexity [15] (right) for blue and white blocks. The Assembly process leads to a measure of structural complexity that accounts for the structure of the object and how it could have been constructed, which is in all cases computable and unambiguous.

**Figure 2 entropy-24-00884-f002:**
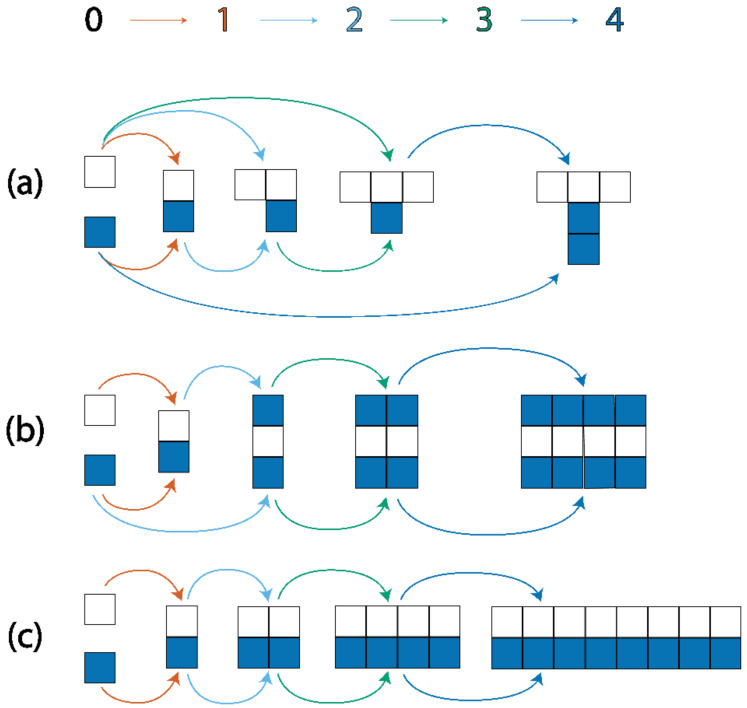
The basic assembly concept is demonstrated here. Each of the final structures can be created from white and blue basic objects in four joining operations, giving an assembly index of 4. Pathway (**a**) shows the creation of a structure that can only be formed in four steps by adding one basic object at a time, while pathway (**c**) represents the maximum increase in size per step, by combining the largest object in the pathway with itself at each stage. Pathway (**b**) is an intermediate case.

**Figure 4 entropy-24-00884-f004:**
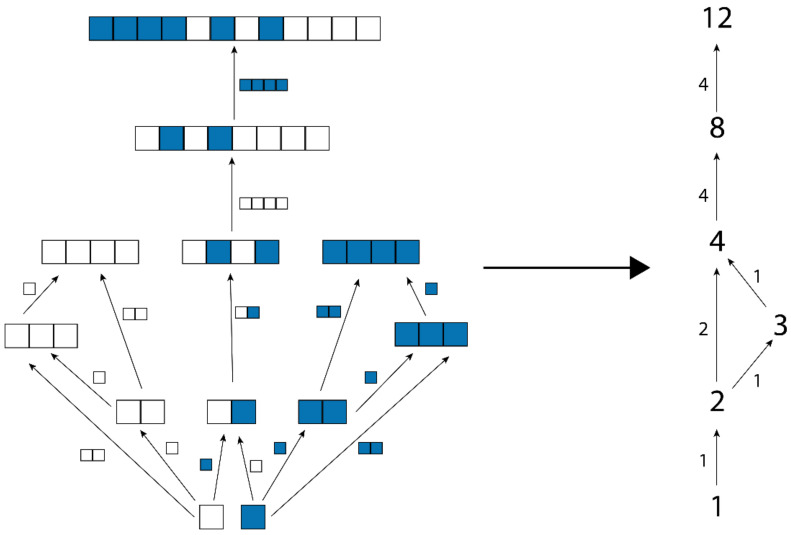
An assembly map that maps an assembly space of white and blue blocks onto integers representing the object size.

**Figure 5 entropy-24-00884-f005:**
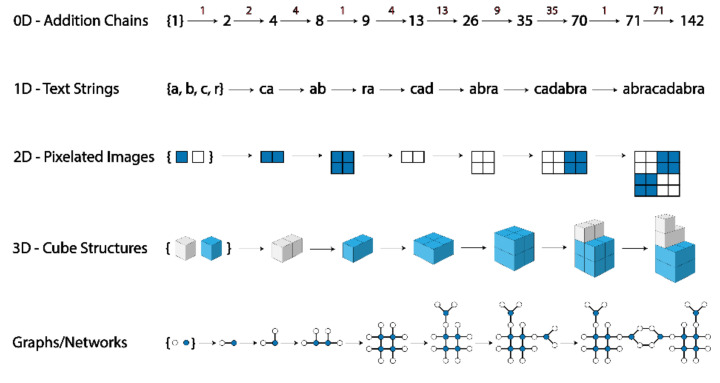
Example assembly pathways for systems of varying dimensionality.

**Figure 6 entropy-24-00884-f006:**
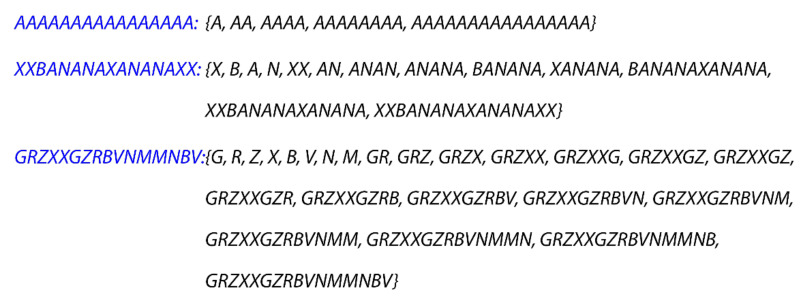
Examples of text assembly pathways for 16-character strings. The first example demonstrates the shortest possible assembly index of any such string. The second example has a nontrivial assembly pathway, while the third example is a string without any shorter pathway than adding one character at a time. This model assumes that text fragments cannot be reversed when concatenating.

**Figure 7 entropy-24-00884-f007:**
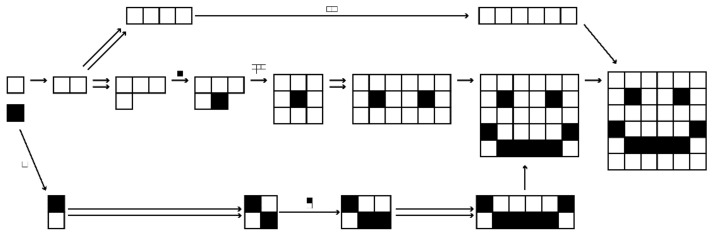
Illustrative assembly pathway of a two-dimensional image. This does not necessarily represent the minimal assembly pathway for this shape. Here, images that are rotated or reflected are considered equivalent.

## Data Availability

Not applicable.

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
