# Peer review of "Formalising the Pathways to Life Using Assembly Spaces"

_entropy, 2022, doi:10.3390/e24070884_

Round 1
Reviewer 1 Report
In this contribution, the authors present the “foundations of a new theoretical approach to agnostically bound the amount of information required to construct an object” (quoted from p. 2 of the manuscript). Generally speaking, having a principled way to quantify the complexity of any composite object is one of the holy grails of complexity theory, and would be an obvious boon for agnostic “evidence of life” measures.
The present manuscript focuses mostly on those said foundations: they are a set of definitions, lemmas, and theorems that imply that the “pathway index” (the author’s measure of complexity) satisfies a number of important requirements, such as transitivity, as well as bounds on the measure. Furthermore, they assert that the measure is computable.
I have no objections to the particular construction, definitions, and proofs shown here. While I find the notation to be somewhat cumbersome (in particular concepts and notations are used in the Results section that are only defined in the Appendix) this is clearly a matter of taste. For example, the word “quiver” is, in my experience, not very common in graph theory, where you would just say “directed graph”. Also, the “graph-theoretical prerequisites” in Appendix A are very particular prerequisites for this particular theory, not (as the title of the section implies) general prerequisites. For this reason, I would prefer to see those definitions in the main section before the results. This would improve readability tremendously, in particular because the notation and vocabulary used are non-standard.
However, there are some comments I would like to make that I think are important enough to raise. For example, while the authors state that they are interested in quantifying the information that is necessary to specify the assembly pathway, they never attempt to do this (here or in any of the earlier papers that I looked at). Intuitively speaking, it is clear that such a measure can be defined, and it should scale with the assembly index. And it turns out that precisely such a measure has been defined already: it is the “thermodynamic depth” of Lloyd & Pagels (Annals of Physics 188, 186 (1988)). This measure introduces a depth D that is a “measure of the amount of information necessary to specify the particular (assembly) trajectory of a complex system”. They even give an example of the assembly of complex proteins.
They also start with a set of axioms, and show that their measure is the only one that satisfies those axioms. It is also computable because it only requires measurements. It is somewhat surprising that the authors were not aware of this contribution, but perhaps it is due to the somewhat unfortunate title of the paper: the measure really has nothing to do with thermodynamics” (even though it can certainly be applied to thermodynamical systems). It really should have been called “Complexity as Information-theoretic Depth”.
Now, one of the things that these authors prove in this paper is that the information necessary to specify the pathway that it takes to assemble the structure is very likely equal to the information content of the “degrees of freedom” of the object (Theorem, p. 199). While that part is fairly opaque, what they seem to imply is that the information necessary to specify the assembly trajectory should be roughly equal to the amount of information necessary to specify the object’s function within the environment, simply by measuring the distribution function of the component abundance. If you think of the components as monomers, this result implies that the amount of information necessary to specify the assembly trajectory of a molecule must be equal to the amount of information within the sequence that specifies the structure (the program). In the case of a biological molecule, that would be the information content of the DNA sequence that codes for the molecule. The latter measure exists: it is the Shannon information content of a biomolecule, which I defined in Ref. [1] below (I’m signing this review).
That reference, incidentally, should be used instead of Ref. [14] (which is just a reference to Shannon’s paper). More to the point, the discussion of the “information content of a string” following that reference (lines 469-474) makes no sense at all. The formula on line 472 is just completely gibberish. In any case, you cannot define the entropy of a single string. But this is not a fatal flaw at all as it is easily fixed: how to do this correctly is described at length in Ref. [1].
Minor comments: It is not clear what “the biological nature of the works of a Shakespearen play” means (line 158). What are the “works of a play”? What are their biological nature?
Line 152: “a system that produces long random strings will generate many that have high assembly index”. I wouldn’t say many: there are “some”, but they are definitely exponentially suppressed (rare). And indeed they will not have high abundance.
Line 53: the authors comment that information in biomolecules may be encrypted “with random keys from the environment”. This is probably true (even though Ref. [9] does not give any examples of this). I’d like to point out that it is much more likely that we will find that information is encrypted using keys that are other molecules in the same organism (rather than in the external environment). This is demonstrated in Ref. [2] where the random variables are sequences of neuronal states, but it holds equally for polymers.
In summary, this paper gives some of the background “foundational theory” for the authors’ assembly theory, which I think more mathematically inclined readers have been missing, but it misses a link to “thermodynamic depth” theory that defines the information necessary to specify an assembly trajectory. This depth D should scale with the assembly index that the authors propose. To make that correspondence precise is probably something that should be done in a separate paper, but it is certainly worth mentioning. Assembly depth should also scale with the information content of the program that generates the molecule (for example, the gene) but that would be another paper still.
Finally, figure 3 is the same as Supp. Fig. 1 of the 2021 paper. Not sure if this creates copyright problems. I do not consider this plagiarism.
[1] C. Adami & N.J. Cerf, Physical complexity of symbolic sequences. Physica D 137 (1995) 62, see also C. Adami, Information theory in molecular biology, Phys. Life Rev. 1 (2004) 3
[2] A. Tehrani & C. Adami, Entropy 22 (2020) 385, see also Bohm et al. Entropy 24 (2022), 735.
Author Response
We would like to thank both reviewers for their positive comments and suggestions on how to improve the manuscript. Following the reviewer feedback, we have made the following changes, which we feel address all the points raised.
In response to comments from both reviewers
- We have moved the Appendix A “Graph Theoretical Prerequisites” section into the main text as section 2.1, replacing the text that formerly summarised those definitions and referenced the appendix. Definitions have been incremented accordingly, as well as section numbers in section 2, and appendix names (with Appendices relabelled B->A and C->B).
In response to comments from reviewer 1:
- We would like to thank the reviewer for pointing out the paper on Thermodynamic Depth. We are planning a paper on the relation between assembly theory and thermodynamics in the near future, which should be an excellent vehicle to discuss relationship to this theory in-depth. In the meantime, we have added a reference to the paper in this manuscript, in the introduction (as reference 18, with other reference numbers incremented accordingly).
- We have replaced the discussion on the information content of a strings with an alternative text, and replaced reference 14 as suggested.
- We have addressed points relating to the biological origin of a Shakespeare play and the production of long random strings.

Reviewer 2 Report
The paper provides a formal grounding for assembly theory, which is a recent approach to quantifying molecular complexity. It also proves a general result that will allow bounds to be put on the assembly index in cases where it can't easily be calculated directly. This is work that needs to be done in developing the theory, and entropy is a suitable venue, so I see no major barriers to publication.
I only have two real criticisms. The first is that the title, "Quantifying the pathways to life using assembly spaces", suggests an analysis of empirical data about the molecular pathways to life, which doesn't match the content of the paper. I would suggest a title along the lines of "Formalising an approach to quantifying the pathways to life using assembly spaces", which focuses on the formal development of the theory rather than its application. (I hope the authors can think of one that gets the same point across while being less awkward than my suggestion.)
My second criticism is that the paper doesn't always take care to define basic terms before they are used, which makes it harder to read than it needs to be. This will be easily fixed, however, by some minor reordering of paragraphs and by moving some material from the appendix into the main text - I give details in my comments below.
Although I make a lot of comments below, none of them are particularly major. I first list the ones that might require some thought to fix, and then give a lot of more minor comments that should help to clarify the paper's contents.
More substantial comments
Abstract: The abstract doesn't really mention the main results, as captured by theorems 2 and 3. I would suggest at least mentioning that the main results are about putting bounds on the assembly index.
Definition 6 (line 233): should this definition include a requirement that B_{\Gamma'} is non-empty? Otherwise any assembly space has the empty assembly space as a rooted assembly subspace, and that seems to break things.
line 237 "It is rooted if it contains a nonempty subset of the basic objects of Γ." - I think you're claiming that this is equivalent to definition 6, but this is not obvious, even if the issue about the non-empty requirement is fixed. Naïvely, it could be that the assembly substance contains a nonempty subset of the basic objects of Γ but also has another basis object that is not a basis object of Gamma, in which case it would not be a rooted assembly subspace. Perhaps this can't happen, but if so that needs a proof.
line 296 "One such map that is generally applicable is the mapping of each object to its size." - it would be helpful to expand this comment out slightly, by first defining the assembly space you're mapping to and then proving that such a map always exists. (I think the space in question is the one whose objects are natural numbers, which has exactly one edge from n to m if n < m, and where the function phi maps an edge n->m to the object m - n.) Spelling this out would be especially helpful for an audience that is probably not familiar with category theory. (I'm assuming the target audience is not familiar with category theory since this is the first time category theory terminology was explicitly mentioned. That being the case, walking through examples explicitly is likely to be helpful.)
proof of theorem 4 (lines 366-372) - this seems to include an unstated assumption that the set x\downarrow is computable. It's finite, but that doesn't immediately imply that it can be computed if the assembly space is infinite, and I suspect one can construct weird/abstruse examples where it's not. Or maybe I'm wrong and there's an obvious way to compute it, in which case that should be stated in the proof. If not, the computability of this set could be added as an extra assumption in the theorem. (It's a pretty mild assumption IMHO.)
Cosmetic / clarification comments:
Introduction section:
My only comment here is the same as for the abstract. The introduction does a great job of introducing the background and main concepts, but it doesn't seem to summarise the results, and it would be helpful to include that.
Section 2:
The paragraph beginning "A quiver is a directed multigraph..." (line 182) needs to be placed before definition 1 instead of after it. This is because this paragraph defines several terms and symbols that are used in definition 1, and the definition is meaningless unless those terms are defined first. I don't think those terms and symbols will be understood by the target audience without giving definitions first, and some readers will simply stop reading if they see jargon they don't understand at this point in the paper.
In the same paragraph it would be helpful to state the meaning of the min and up arrow operators instead of just referring to the appendix. These operators are used in definition 1, and if you don't say what they mean you force the reader to read the appendix before they can even understand the first definition in the paper, which is silly and defeats the purpose of having an appendix in the first place.
For the same reason, in definition 1 it would help a lot for readability if you would spell points 1-3 out in words as well as symbols. (I note that this is actually done a couple of paragraphs later; perhaps it would be enough just to say something like "first we state the formal definition and then spell out its meaning more informally", so that the reader knows this is coming.)
lemma 1 (line 197) the meaning of the notation "e ~ [ba]" has not been mentioned before this point. (It's explained in the appendix, but one shouldn't assume the reader has read the appendix before the main text.) It's not notation that I've encountered before.
proof of lemma 1 (line 199): "Since ? is an assembly space, we have ?(?) ≤ ?" I suggest adding something like "because there is an arrow from ?(?) to ? by point 3 of definition 1".
line 210 "This means that ..." - why does it mean that, exactly? It doesn't seem obvious.
line 212 "They may use objects that are considered identical (e.g. the same string) but these are separate objects within the space" - I can't figure out what this sentence is supposed to mean, especially the second half of it.
line 214 "Since we can define an assembly map..." - this sentence uses several terms that haven't been introduced yet. It should be rephrased to avoid them, or moved to later in the paper after those terms have been defined.
Definition 4 (line 221) - this definition repeats exactly what was said at the start of the previous paragraph.
line 230 - the vertical bar notation for the restriction of a function's domain is likely to be unfamiliar to a lot of readers and it's not something that can be googled - it would be well worth stating its meaning explicitly.
Definition 7, line 265: the sentence beginning "The augmented cardinality of the (?, ?)" doesn't quite parse grammatically. I guess you mean to say "The augmented cardinality of an assembly space (?, ?) with basis..."
In the same definition: I suggest making "augmented cardinality" its own separate definition, mostly because it depends on the basis and not just on the assembly space.
"Augmented cardinality" seems an odd choice of name, given that it's strictly less than the cardinality. Would "cardinality relative to basis B_\Gamma" make sense? That also makes it clear that it depends on the basis.
Definition 9 (line 290) - it would be helpful to mention the section in the appendix where quiver morphisms are defined.
line 307 "...the essential point being that the image of an assembly space under an assembly map is an assembly space." - I believe this should say "...the essential point being that the image of an assembly space under an assembly map is an assembly subspace."
line 317, the sentence beginning "Essentially" - the "usually" threw me off a bit, so I suggest a rephrasing: "Essentially, since the assembly subspace may have fewer edges, there are in general fewer “shortcuts” for assembling a given object."
lines 326-327 "by the transitivity of subset inclusion (lemma 2)" - I think this should say "by the transitivity of rooted assembly subspace inclusion (lemma 2)".
Proof of theorem 2 (lines 339-347) - it would be helpful to include some paragraph breaks and/or displayed equations here, as it's currently an impenetrable wall of symbols.
line 362: this should specify that the computability result only applies to finite objects.
Discussion section:
lines 414-417 - here you talk about "the" assembly space of positive integers, but you didn't uniquely specify an assembly space. I think you probably want to talk about the assembly space I mentioned above in my comment on line 296. (The thing that's missing is to say there is exactly one edge from x to y whenever y > x.)
lines 428/429 "Assembly spaces can generally be mapped..." - this is a nitpick, but this should probably say "assembly spaces that occur in practice can often be mapped...", since it's easy to think of examples where there is no meaningful notion of size, or where the natural notion of size is not an integer.
lines 482-489 (discussion of Kolmogorov complexity) - it should probably be mentioned around here that Kolmogorov complexity and the assembly index are different in nature and measure rather different things. Kolmogorov complexity only applies to infinite strings while the assembly index only applies to finite ones. Furthermore, a string consisting of the binary expansion of pi has a low Kolmogorov complexity (it only needs a short program to generate it) but I would expect the assembly index of the first n digits of that string to grow at the same rate as for a random string. This is because although there is a pattern it is not in a form that allows the assembly process to take advantage of it.
Figure 7: I think some lines are missing in the small images that serve as labels on the arrows. At least on my screen some of the small white squares are not visible.
line 519, "Assembly theory as described here currently has no simple extension to continuous objects..." - this seems a rather odd claim to me! You never restricted your quivers to have a finite set of objects or a finite set of edges, and up to this point I had assumed that this was because you wanted to include continuous objects. I didn't see anything so far that prevented them from being considered. For example, in your addition chains example you could replace the positive natural numbers with the positive reals and it would still define an assembly space, though one with arguably less interesting properties.
I suppose what you don't currently have is a way to extend the assembly index to the case where the process of constructing an object is continuous, as opposed to requiring a discrete sequence of steps.
Conclusions section:
None of the points in this section were made in the paper or directly supported by it. I think they have been made in previous papers by the same authors though, so a few references would fix that issue.
Appendix C:
Although it's not common practice, I wonder if you would consider repeating the statements of the theorems before their proofs in this appendix? When a paper is written in this style I always have to open two copies of the paper so that I can refer back to the theorem statements, and it seems to me that it would be kinder to the reader if this wasn't necessary.
In the case of theorem 1 (line 769), the main issue was remembering whether the type of f was \Gamma\to\Delta rather than \Delta\to\Gamma. If you prefer not to re-state the theorems, an alternative would be to include this information in the wording of the proof, e.g. "Since $f\colon\Gamma\to\Delta$ is a quiver morphism..."
Formatting issue, line 772 and lines 789-805: all the math on these lines is in bold for some reason.
typos
line 175: "exists and edge" -> "exists an edge"
line 294 "An assembly map is essentially mapping" -> "...is essentially a mapping"
line 316 "in assembly space Γ" -> "in an assembly space Γ"
line 362 "However importantly the assembly index is computable" -> "However, importantly, ..."
line 498 "? ~ [?? ]" - there is an extra space after the 'a'
line 579 "assembly theory might be enable us" -> "assembly theory might enable us"
line 800 ", then there exists" -> "there exists"
Round 2
Reviewer 2 Report
All the issues I found with this paper were minor, and I believe they have been addressed satisfactorily.
Some very minor issues that could be addressed by the authors without another round of reviews:
* the definition of the space of positive integers under addition (section 3.1) is still not quite correct. You have to specify that there is exactly one edge e~[zx] if z>x, and no edges otherwise. Without that the definition doesn't uniquely specify an assembly space.
* the definition of the space of positive integers under addition is now referred to in definition 19 - a "see section 3.1" would be helpful there.